# Reliability and Validity of the Malaysian English Version of the Diagnostic Criteria for Temporomandibular Disorder (M-English DC/TMD)

**DOI:** 10.3390/healthcare10020329

**Published:** 2022-02-09

**Authors:** Farah Nur Tedin Ng, Kathreena Kadir, Zamros Yuzadi Mohd Yusof

**Affiliations:** 1Department of Oral & Maxillofacial Clinical Sciences, Faculty of Dentistry, Universiti Malaya, Kuala Lumpur 50603, Malaysia; abc_3012@yahoo.com; 2Department of Community Oral Health and Clinical Prevention, Faculty of Dentistry, Universiti Malaya, Kuala Lumpur 50603, Malaysia; zamros@um.edu.my

**Keywords:** M-English DC/TMD, cross-cultural adaptation, reliability, validity, temporomandibular disorders

## Abstract

This study aimed to assess the reliability and validity of Graded Chronic Pain Scale 2.0 (GCPS 2.0) and Jaw Functional Limitation Scale-20 (JFLS-20) of the Malaysian English version of Diagnostic Criteria for Temporomandibular Disorders (M-English DC/TMD). GCPS 2.0 and JFLS-20 underwent psychometric analysis involving content, face, criterion, and construct (exploratory factor analysis (EFA), convergent, discriminant, known group) validity including internal and test-retest reliability on 208 samples. The construct validity was assessed against 14 hypotheses and non-parametric statistics were used to assess the data. The GCPS 2.0 and JFLS-20 had high internal consistencies (α = 0.85 and 0.96) with an intraclass correlation coefficient value of 0.95 and 0.97, respectively. The content validity index for GCPS 2.0 and JFLS-20 were 0.87 and 0.95, respectively. The EFA of GCPS 2.0 identified one factor whereas three factors were identified for JFLS-20. Both instruments had moderate to strong positive correlation with other instruments when assessing for concurrent (*r* = 0.75–0.80, *p* < 0.01) and convergent (*r* = 0.58–0.70, *p* < 0.01) validity, while moderate to high negative correlation (*r* = −0.86–−0.68, *p* < 0.01) against the global self-rating oral health items. Based on the study, GCPS 2.0 and JFLS-20 of the M-English DC/TMD proven to be reliable and valid for use in the Malaysian population with TMD.

## 1. Introduction

Temporomandibular disorders (TMD) are a heterogenous group of conditions affecting the temporomandibular joints (TMJ), masticatory muscles, and surrounding structures [1]. It affects between 5–12% of the general population with the incidence peaks between 20 to 40 years of age; twice as common in women than in men [1]. In addition to orofacial pain and headaches, TMD signs and symptoms include TMJ noises and locking, limited or abnormal jaw movements, and otologic complaints like the feeling of ear fullness, tinnitus, vertigo, and hearing loss [1,2]. Among many factors that had been recognised to be the factors contributing to the onset and persistence of TMD are psychological and emotional distress, parafunctional habits, acute stress to the jaw, maxillofacial instability, joint laxity, other underlying rheumatic or musculoskeletal disorders, poor general health, and unhealthy lifestyles [3,4].

The Diagnostic Criteria for Temporomandibular Disorders (DC/TMD) was developed based on evidence-based criteria and has been a diagnostic protocol for TMD research. It is being used as an internationally standardised tool in the screening and diagnosis of TMD [5]. DC/TMD is a dual-axis instrument, whereby Axis I contributes to physical diagnosis which consists of TMD pain screener, demographic, symptom questionnaire, and physical examination. Axis II assesses the psychosocial status of a patient which consists of pain drawing, graded chronic pain scale (GCPS) version 2, jaw function limitation scale (JFLS)-8, JFLS-20, patient health questionnaire (PHQ)-4, PHQ-9, PHQ-15, generalised anxiety disorder (GAD)-7, and the oral behaviour checklist (OBC). These validated instruments facilitate screening and confirmation of patients with a range of simple to complex TMD presentations [5].

With the advancement in research seen at present, any instrument or measurement tool that has been developed or that has to be adapted to a population has to undergo a process of cross-cultural adaptation and psychometric analysis to prove that it is valid and reliable and if it represents the population in question. Validation assesses a tool whether it is measuring what it is supposed to measure [6] whereas reliability assesses whether the tool is stable and whether it can reproduce the same results if used again in a constant condition [7]. DC/TMD itself had undergone multiple processes of translation and cultural adaptation by numerous countries worldwide and was subjected to psychometric evaluation [8]. At present, the Malay version of DC/TMD is already available for download from the International Network for Orofacial Pain and Related Disorders Methodology (INfORM) official site [9]. The same instrument had undergone cross-cultural adaptation to the Malaysian population followed by a complete validation process [10].

Malaysia is globally known as a multiracial and multilingual country where English is placed as the second language [11]. The language became more significant when the government came up with plans for the nation’s future development. These plans involved Malaysia’s growth into an industrialised nation in line with vision 2020 [12]; with the establishment of the multimedia super corridor (MSC) [13], as well as the establishment of Malaysia as a regional centre of education [14]. The plans have implications for a change in the language policy demanding self-improvement in the citizens’ language competence allowing English to play a more dominant role [15]. Hence, a questionnaire in the Malay language without an English translation can be said to be incomplete. This study aimed to assess the reliability and validity of graded chronic pain scale 2.0 (GCPS 2.0) and jaw functional limitation scale-20 (JFLS-20) of the Malaysian English version of DC/TMD (M-English DC/TMD) in view of English as Malaysian’s second language as both instruments are used routinely in patient management of TMD.

## 2. Methods

This study comprised of two phases: (a) Phase I: cross-cultural adaptation and theoretical construct validation, focusing on the sensibility of the measure, its comprehensibility, its content and face validity, its replicability, and the suitability of the scales [16], followed by (b) Phase II: psychometric evaluation consisting of reliability and empirical construct validation of the instruments.

Data collection was conducted at the Faculty of Dentistry, Universiti Malaya from November 2018 to June 2020. It was carried out in accordance with the 1964 Declaration of Helsinki. Ethical approval for the study was granted by the Medical Ethics Committee, Faculty of Dentistry, Universiti Malaya [Reference: DF OS1820/0073P). The study was financially supported by Universiti Malaya Dental Postgraduate Research Grant (Reference: DPRG/23/19). Written informed consent was obtained from all the respondents before data collection.

### 2.1. Phase I: Cross-Cultural Adaptation and Theoretical Construct Validation

Cross-cultural adaptation of an instrument or measure has two key components which are (1) translation and (2) adaptation [17]. The English DC/TMD in the present study was only subjected to the adaptation process as it was adapted for English-speaking Malaysian population. Since no changes were made to the language, the adaptation process was conducted with regard to the idiom, and the cultural context and lifestyle of the targeted population. Validation according to a theoretical construct uses a group of individuals which then assess whether the questionnaire represents the construct of interest. It is also called translational representational validity. This phase started with the selection of six experts consisting of a group of specialists from the oral and maxillofacial surgery discipline, oral pathology and oral medicine, orthodontics, prosthodontics, and psychology unit to assess the content validity of the M-English DC/TMD Axis II using a preformed assessment form that allowed the experts to comment and rate each item in all the Axis II instruments using a 4-point ordinal rating scale where 1 = not relevant, 2 = somewhat relevant, 3 = quite relevant, and 4 = very relevant. The content validity index (CVI) for each item, I-CVI (Number of Agreements/Number of Raters) in all the instruments followed by the CVI for the whole scale, S-CVI (Number of Items with Agreement/ Number of Items) were then calculated based on Lynn’s method [18]. Comments by the experts were also reviewed by the research members to address their opinions in case any of the items were irrelevant or unsuitable for the Malaysian population.

In addition to content validity, the theoretical construct validity was further evaluated with the establishment of face validity of the M-English DC/TMD by a face-to-face interview with 10 adult respondents while they filled up the questionnaire. After filling up each item, the respondents were asked to describe what they have understood; produce an alternative or to rephrase the item if there was any obscurity; and give an overall comment on the whole questionnaire. Obscurities detected were also compiled to be reviewed. Changes were then made in a discussion among the research team members.

### 2.2. Phase II: Reliability & Empirical Construct Validation

For psychometric analysis, we chose two instruments of the DC/TMD: (i) the graded chronic pain scale 2.0 (GCPS 2.0), and (ii) the jaw functional limitation scale 20-item version (JFLS-20) for reliability and validity assessment. These instruments were chosen as they represent two of the core outcomes measures for chronic pain (pain and physical functioning, respectively) following the Initiative on Methods, Measurement, and Pain Assessment in Clinic Trials (IMMPACT) guidelines [5,19]. Phase II involved the assessment of reliability (internal consistency and test-retest reliability) and validity (concurrent validity, and construct (factorial, convergent, discriminant, known-group)) of the two instruments.

To assess the psychometric properties, we recruited adults with the following inclusion criteria: aged 18 years and above and able to comprehend and write in English language. Those with recent trauma or organic pathology related to the TMJ region such as tumour, medically diagnosed with psychiatric disorders, and those unable to provide consent were excluded. For sample selection, a mixture of TMD patients and the general public who fulfilled the inclusion and exclusion criteria were recruited into the study. TMD patients were selected among patients who were referred and attended the TMD multi-disciplinary team (MDT) combined clinic at the Faculty of Dentistry, University of Malaya. These patients were diagnosed using the original DC/TMD instrument applied by the MDT of the TMD combined clinic. Individuals without TMJ pain were selected among students and staff of the university.

The sample size calculation was based on the respondents-to-item ratio which was 5 to 10 respondents per item of each questionnaire [20]. Overall, a total of 208 respondents were involved for the first convenient sample, consisting of 119 general public individuals and 89 TMD patients. Consequently, a second convenient sample of 50 individuals (25 non-TMD individuals and 25 TMD patients) from the initial 208 respondents were asked to fill up the same questionnaire after 10–14 days to assess for test-retest reliability [21,22]. In this study, the TMD patients were identified based on a questionnaire, physical examination, and decision tree in Axis 1 of DC/TMD.

### 2.3. Study Tools

The GCPS 2.0 consists of 8 items with three subdomains: (i) “Characteristic Pain Intensity” (items 2 to 4), (ii) “Interference of Activities” (items 6 to 8), and (iii) “Days with Interference” (item 5). Item 1 (*On how many days have you felt pain in the past 6 months?*) has demonstrated a very wide range of scores and has often been excluded in statistical analyses especially in the determination of internal consistency [23].

The JFLS-20 consists of 20 items with three functional limitation subdomains: (i) “Mastication” (items 1 to 6), (ii) “Vertical Jaw Mobility” (items 7 to 10), and (iii) “Verbal and Non-Verbal Communication” (items 11 to 20). The response options for each item ranged from 0 (no limitation) to 10 (severe limitation).

In this study, we used the short version of the brief pain inventory (S-BPI) to assess concurrent validity for the GCPS 2.0, and the oral health impact profile for TMD (OHIP-TMD) for the JFLS-20. The BPI consists of 9 items with two subdomains which are found to be similar to the GCPS 2.0. The BPI “Pain severity” and “Pain interference” subdomains are comparable to the GCPS 2.0 “Characteristic pain intensity” and “Interference of Activity” subdomains, respectively.

The oral health impact profile for TMD (OHIP-TMD) consists of 22 items with 7 subdomains which are the “Functional limitation” (items 1 to 2), “Physical pain” (items 3 to 7), “Psychological discomfort” (items 8 to 11), “Physical disability” (items 12 to 13), “Psychological disability” (items 14 to 18), “Social disability” (items 19 to 20) and “Handicap” (items 21 to 22). Each OHIP-TMD item was scored on a 5-point Likert scale (never, hardly ever, occasionally, fairly often, and very often) [24]. The OHIP-TMD detects the changes in the impact of TMD on quality of life and hence has a well-established association with symptoms and functional limitation in TMD [25].

The global self-report oral health (GSROH) status consists of a single question “*In your opinion, how do you rate your oral/jaw health status?*” [26] and responses were scored on an ordinal scale ranging from 1 (very poor) to 5 (very good). A lower GSROH score indicates a poorer oral health profile.

### 2.4. Statistical Analysis

The collected data were analysed using the IBM SPSS Statistics for Windows, version21 (IBM, Armonk, NY, USA). The demographic data were analysed using descriptive statistics to determine the respondents’ characteristics such as age, gender, race, marital status, highest education level, and income.

The internal consistency reliability of the GCPS 2.0 and JFLS-20 was assessed to determine their homogeneity by calculating the inter-item correlation, Cronbach’s alpha, and Cronbach’s alpha if item deleted. In terms of Cronbach’s alpha, a value ≥0.7 is considered good reliability [27]. The Cronbach’s alpha values were determined using total scores of the GCPS 2.0 and JFLS-20. The Cronbach’s alpha if item deleted, inter-item correlation, and item-total correlation were also determined for both instruments. The test-retest reliability was assessed by determining the intraclass correlation coefficient (ICC) where a value of ≥0.7 is considered good test-retest reliability. For this purpose, 50 individuals were selected to answer the questionnaire again after 10 to 14 days.

Factorial validation assesses the construct by the use of a statistical model known as the factorial analysis by which the EFA of principle components of the GCPS 2.0 and JFL-20 with Oblimin rotation method was established in this research to identify the number of factors contributing to each instrument. We conducted EFA to explore if the factors yielded when tested in the Malaysian population with a different language and sociocultural settings are similar to the factors contained in the original instruments. This is similar to past studies done on the validation of the instruments. The factor loadings were examined to confirm the items being loaded to each subdomain. An eigenvalue of 1 was set as the limit of variance for each factor. We hypothesised that “Characteristic Pain Intensity” and “Interference of activity” are the two factors influencing the total score of the GCPS 2.0 and that “Mastication”, “Vertical jaw mobility” and “Verbal & non-verbal communication” are the three factors influencing the total score of the JFLS-20.

In this study, for concurrent validity which measures whether both instruments are comparable to other available instruments, we computed the correlation between GCPS 2.0 scores with BPI scores and JFLS-20 scores with mean OHIP-TMD scores using the Spearman’s rho correlation test. Construct validity determines how accurate the instruments measure the construct(s) they are supposed to measure. In terms of convergent validity, we determined the correlation between GCPS 2.0 scores and scores of the “physical pain”, “psychological discomfort”, “physical disability”, “psychological disability”, “social disability” and “handicap” subdomains of the OHIP-TMD; as well as between JFLS-20 scores and scores of the “functional limitation” subdomain of OHIP-TMD using the Spearman’s rho correlation test to assess for convergent validity. We hypothesised that respondents with physical pain, psychological discomfort, physical disability, psychological disability, social disability, and handicap would have higher GCPS 2.0 scores than those with no physical pain whereas respondents with jaw functional limitation would have higher JFLS-20 scores than those with no jaw functional limitation.

Spearmen’s rho correlations between GCPS 2.0 and JFLS-20 scores with the 1-item GSROH scores were computed to assess discriminant validity. Subjects who perceive their jaw health as very poor indicate the presence of jaw pathology and tend to have pain and functional limitation than subjects who perceive their jaw health as very good. We hypothesised that respondents with a better perception of their oral or jaw health status would have lower GCPS 2.0 as well as JFLS-20 scores than those with a poorer perception of their oral or jaw health status. The known-group validity was established by comparing GCPS 2.0 and JFLS-20 scores of the general public individuals and TMD patients through a Mann-Whitney U test. We hypothesised that respondents with TMD would have higher GCPS 2.0 and JFLS-20 scores than those with no TMD.

Finally, hypothesis-testing validity was assessed by evaluating the hypotheses based on the results. Fourteen hypotheses were generated at the beginning of our research.

## 3. Results

### 3.1. Sample Size and Characteristics

A total of 208 respondents (67 male, 141 female) participated in the study, aged between 18–60 years and above. More than half (57.7%) of the sample were Malays with 53.8% aged 18–30 years and 48.6% were never married. All of them were literate and 74.1% of them have received tertiary level of education. In general, they were mostly from the middle-income group. The sample characteristics are summarized in Table 1.

### 3.2. Validity

I-CVI value of the GCPS 2.0 item 1 and JFLS-20 item 17 were lower than the recommended cut-off value of 0.78 [28]. However, the S-CVI of both instruments was still above the recommended cut-off value of 0.80 [28]. Table 2 demonstrates the changes made following content and face validity of the M-English DC/TMD.

EFA of the principal components of the GCPS 2.0 with Oblimin rotation method identified one factor with seven items (factor loading: 0.681–0.918) corresponding to an eigenvalue of 6.07. The Kaiser-Meyer-Olkin measure of sampling adequacy of 0.88 was higher than the recommended value of 0.60, and the results of Bartlett’s test were significant (chi-squared = 2179.59, df = 28, *p* < 0.001), indicating the adequacy of the data for performing principal components analysis. A scree plot was drawn to support the 1-factor structure (Figure 1), which explained the total variance of 75.92%.

On the other hand, EFA of the principal components of the JFLS-20 with Oblimin rotation method showed that the Kaiser-Meyer-Olkin coefficient was good (0.85), indicating sample adequacy, and Bartlett’s test of sphericity was significant (Chi-square= 6718.92, df = 190; *p* < 0.001), confirming the adequacy of the data for factor analysis. A 3-factor structure; “Mastication” (item 1–5, 7–8, 12, 14–15, 20) with factor loading 0.412–0.995, “Vertical jaw mobility” (item 9–10, 13) with factor loading 0.430–0.543, and “Verbal and non-verbal communication” (item 6, 11, 16–9) with factor loading 0.591–0.964 were identified with an explained total variance of 80.54% supported by a scree plot (Figure 2).

The assessment of concurrent validity yielded positive associations when a Spearman’s rho correlation test was done between the GCPS 2.0 and BPI subdomains, and the JFLS-20 with OHIP-TMD subdomains as shown in Table 3. Convergent validity testing resulted in a moderate positive correlation between the GCPS 2.0 and JFLS-20 with all the OHIP-TMD subdomains whereas discriminant validity testing resulted in a moderate negative correlation with the GSROH (Table 4). Known-group validity determination in Table 5 showed that TMD patients had significantly higher mean ranks for both GCPS 2.0 and JFLS-20 scores than the general public individuals. At the end of the study, 13 out of 14 of the constructed hypotheses were in accordance with our initial prediction. Construct validity assessment resulted in 1 out of 14 hypotheses being rejected (92.86% in accordance with the predicted outcome).

### 3.3. Internal Consistency and Test-Retest Reliability

The internal consistency was assessed by calculating the Cronbach’s alpha for the GCPS 2.0 and JFLS-20, which yielded high internal consistency with an α value of 0.85 and 0.96, respectively. The corrected item-total correlation values for the GCPS 2.0 and JFLS-20 ranged from 0.70–0.89 and 0.55–0.91 respectively. All the items of GCPS 2.0 and JFLS-20 surpassed the value of 0.60 which is the least recommended correlation value except item 19 in the JFLS-20 [27,29]. Generally, the α value of the GCPS 2.0 and JFLS-20 was not significantly reduced if any of the items were removed. The inter-item correlation for the GCPS 2.0 items ranged from 0.57–0.96 with the lowest correlation between items 4 and 5 (0.57) and the highest correlation between items 6 and 7 (0.96). On the other hand, the inter-item correlation of JFLS-20 items ranged from 0.31–0.98. The lowest is the correlation between items 6 and 19 (0.31) while the highest is between items 1 and 2 as well as between items 16 and 17 (0.98).

Test-retest reliability was determined by selecting 50 respondents consisting of 25 TMD and 25 general public individuals who answered the same questionnaire after 10–14 days. Superior agreements were demonstrated by both GCPS 2.0 and JFLS-20 indicating excellent stability as shown in Table 6.

## 4. Discussion

The quality of the adaptation process was assessed based on the instrument’s sensibility which encompassed the purpose of the measure, its comprehensibility, its content and face validity, its replicability, and suitability of the scales [17]. Furthermore, an instrument adapted into another culture must also have a similar performance to its original version [21].

In this research, we started with a content validity assessment based on comments and ratings by a group of six experts. The experts had given a low rating for item 11 in PHQ-15 (item *Pain or problems during sexual intercourse*) where 3 out of 6 experts rated it as irrelevant as the item has low relevance with TMD and was incompatible with the conservative culture of Malaysians. Some of the food items in the JFLS-8 and JFLS-20 as well as the musical instruments in the OBC were also found to be less common. Subsequently, the research team members came to a decision to keep item 11 in PHQ-15 as it fits within the TMD biopsychosocial illness model and other changes were made to suit the Malaysian culture (Table 2).

In terms of reliability and validity, the GCPS 2.0 and the JFLS-20 have Cronbach’s alpha values above 0.70 which indicated high internal consistencies that were comparable with other studies. The GCPS 2.0 achieved comparable internal consistency with the Spanish [23], Italian [30], German [31], and the original English [32,33] versions with α values ranging between 0.82 and 0.91. All item-total correlation values for the GCPS 2.0 scored ≥0.70 indicating good internal consistency. On the other hand, the JFLS-20 demonstrated an α value of 0.96 comparable to the original English version [32,33] and the Chinese version [34] which scored 0.95 and 0.91, respectively. All the item-total correlation values of the JFLS-20 were above the satisfactory value of 0.60 except for item 19 (“*kiss*” item). This may be due to a slight difference in social upbringings of Malaysians whereby most Malaysians kiss their family members relatively less than those from other countries where kissing is to greet others upon meeting. Shaking hands is more common in Malaysia. Hence, this item may be slightly foreign to Malaysians however it is still relevant.

The test-retest reliability measures how similar the results are when measured at two different settings in which the consistency of measurement repetition is assessed. The recommended time span from 10 to 14 days was used in the present study [35]. In this study, the ICC values for the GCPS 2.0 and JFLS-20 were 0.95 and 0.97 respectively, which are close to 1 indicating excellent stability. The ICC value of the GCPS 2.0 was slightly higher compared to the Spanish [23] and German [35] versions (ICC value of 0.81 and 0.92 respectively). This could be due to the larger number of respondents involved (*n* = 50) in our test-retest reliability assessment compared to theirs (*n* = 27 and *n* = 46 respectively). Similarly, the JFLS-20 scored an ICC value much higher than the Chinese version [34] that scored a value of 0.87 (*n* = 60). The significant difference could be due to the different statistical methods of analysing the ICC whereby we used type A intraclass correlation coefficients using an absolute agreement definition while the other studies use the Pearson’s correlation coefficient test.

Content validity is assessed through a qualitative (experts ‘comments) and quantitative approach (CVI calculation) [21]. According to Lynn (1986), I-CVI must be 1.00 in a group of five or fewer experts and must not be any lower than 0.78 in a group of six or more experts. Based on the ratings by six experts, all the items in the GCPS scored an I-CVI value above 0.78 except for item 1 *“how many days in the last 6 months have you had facial pain*?*”*. The two experts who disagreed commented that the 6 months duration was too long, and patients may not recall the number of days accurately. Our research team members had decided to retain the item as chronic pain is defined as pain that presents for more than half the days in the prior 6 months [36]. All the items in the JFLS-20 also scored an I-CVI value of more than 0.87 except item 17 “*Frown*”. This finding was also demonstrated by the JFLS-20 of the Chinese version [34] which scored the lowest I-CVI value (0.72) for item 17. However, the JFLS-20 in their study was still considered as valid based on the S-CVI.

The GCPS 2.0 subdomains in the present study were correlated with subdomains of the Brief Pain Inventory (BPI) questionnaire for concurrent validity determination and were found to have a moderate to strong positive correlation. The BPI is a gold standard in assessing pain especially cancer pain and has been widely used for assessing pain control using medication. Its subdomains measuring pain severity and quality of life based on life interference are comparable to the subdomains of the German GCPS [35]. Previous researchers had also used BPI to assess pain among TMD patients [37]. On the other hand, the JFLS-20 total score was correlated with the OHIP-TMD to determine its concurrent validity and was found to have moderate positive correlation. The OHIP-TMD has been proven to be valid and reliable in addition to being short, easy and less time-consuming [24]. It has been cross-culturally adapted in China and was proven to be valid and reliable for their TMD patients [38]. The present study uses the Spearman’s rho correlation test as the data were ordinal data and not normally distributed as suggested by Vetter T.R. and Schober P. [39] and we found that both instruments were concurrently valid as their correlation with the external criterion were above the recommended value of 0.70 [22].

Convergent validity was tested through the correlation between the GCPS 2.0 and JFLS-20 with the OHIP-TMD questionnaire. The OHIP-TMD has similar constructs with the two instruments which are the physical pain, psychological discomfort, physical disability, psychological disability, social disability, and handicap subdomains [25]. Thus, a high correlation was expected between the instruments and we hypothesised that respondents who have any of those constructs will score higher on the GCPS 2.0 and JFLS-20.

Discriminant validity was established by correlating the two instruments with another instrument that was predicted to result in an opposite outcome. The most common complaint from TMD patients will be pain followed by a disturbance in jaw function. Hence, most TMD patients would regard themselves as having a poorer oral and jaw health status than those without TMD. We hypothesised that those who have a better perception of their oral or jaw status will have lower GCPS 2.0 and JFLS-20 scores. Hence, the GSROH status was incorporated in our study to assess the perception of the respondents regarding their oral or jaw status.

Known-group validity was determined to differentiate between TMD patients and general public individuals. We hypothesised that TMD patients would have higher GCPS 2.0 and JFLS-20 scores than those who do not have TMD. This study proved that TMD patients exhibited higher scores compared to non-TMD respondents. In another study, the JFLS-20 was tested among 5 diagnostic groups (TMD, burning mouth syndrome, primary Sjögren syndrome, skeletal malocclusion, and healthy controls) in which TMD patients reported a significantly more limitation than other groups [33].

Factorial validity was verified through EFA whereby the test allowed us to identify the number of factors in which the variables were strongly related to each other within each instrument [20]. Previous studies as in the validation of the German version of the GCPS [31], the validation of the Italian version concerning musculoskeletal disorders [30], and the validation of the Portuguese version [40], the GCPS demonstrated two factors which were the “characteristic pain intensity” and “disability score”. Hence, we hypothesized that “characteristic pain intensity” and “interference of activity (disability)” are the two factors influencing the total score of the GCPS 2.0. However, in our study, only one factor was identified. This finding is similar to the finding in the United Kingdom and Turkish versions, as well as the original first GCPS whereby only one factor was identified and the instrument was deemed as unidimensional [36,41,42]. The previous researchers claimed that since the GCPS 2.0 was developed to measure pain, it is reasonable that only one factor represented most of the variance among TMD patients. Therefore, our hypothesis *“characteristic pain intensity and interference of activity are the two factors influencing the total score of the GCPS 2.0”* was rejected.

On the other hand, according to the original JFLS-20 [33] and its Chinese version [34], three factors were reported in their studies. We too hypothesized that “mastication”, “vertical jaw mobility”, and “verbal and non-verbal communication” are the three factors influencing the total score of the JFLS-20. In our study, three factors were extracted through EFA. However, the factor loading of the items was not in accordance with the previous studies. The first factor (mastication) which accounted for 62.12% of the total variance consisted of items 1, 2, 3, 4, 5, 7, 8, 12, 14, 15, and 20. Here, the items “*Open wide enough to bite from a whole apple*” (item 7) and “*Open wide enough to bite from a whole sandwich*” (item 8) were loaded into the “mastication” construct probably because the item themselves consist of food items (the word “*apple*” and “*sandwich*”) which triggered a memory of eating the food instead of just the act of opening the mouth to begin eating. However, the factor loading of items “*Yawn*” (item 12), “*Sing*” (item 14) and “*Laugh*” (item 20) into the “mastication” construct may be explained by the pain felt while performing any kind of jaw function in TMD patients experiencing pain whereas item “*Putting on a happy face*” (item 15) may be misinterpreted as feeling happy emotionally rather than the act of it. TMD patients with chronic pain and jaw function interference would have some dissatisfaction leading to emotional disturbance in their daily lives [43]. Hence, this effect was projected in our study. The second factor (verbal and non-verbal communication) represented 12.48% of the total variance and was consisted of items 6, 11, 16, 17, 18, 19. Somehow the item “*Eat soft food requiring no chewing*” (item 6) was loaded into the “verbal & non-verbal communication subdomain and not the “vertical jaw mobility” subdomain. We would like to argue that item 6 may not require much activity of the mandible and thus did not load into the “vertical jaw mobility” construct but had inversely demonstrated the potential characteristics of “verbal and non-verbal communication” instead. Lastly, the third factor (vertical jaw mobility) which accounted for 5.94% of the total variance was consisted of items 9, 10, and 13. Here, the item “*Talk*” (item 13) may be considered the same as item “*Open wide enough to talk*” (item 9) by the respondents hence loading into the “vertical jaw mobility” construct.

In the present study, a total of fourteen specific hypotheses were predicted for both the GCPS 2.0 (ten hypotheses) and JFLS-20 (four hypotheses). Based on the results obtained from the determination of convergent, discriminant, known-group, and factorial validity, we found that 13 out of 14 of the constructed hypotheses were in accordance with our initial prediction. As elaborated by Terwee et al. in 2007, a positive rating of construct validity is given if the predicted hypotheses were specific and a minimum of 75% of the results were in accordance with the predicted hypotheses. Hence, we can conclude that the GCPS 2.0 and JFLS-20 were valid. In summary, results gained in the present study have confirmed the reliability and validity of the M-English DC/TMD. The validated M-English DC/TMD will serve as the first standardized, up-to-date and conclusive tool to aid in the diagnosis of TMD among Malaysians. Further detailed studies on TMD will be much more feasible once a standardized tool is made available to researchers throughout the country.

The present study has several limitations whereby the respondents were not recruited randomly, and this might have some effect on the research findings as the study may not include samples from all social backgrounds. Secondly, this study is a single-centred study whereby the group of respondents involved may not accurately represent the whole population of Malaysia that consists of a multiracial and multi-ethnic community.

## 5. Conclusions

In accordance with the results derived in this study, we can conclude that the GCPS 2.0 and JFLS-20 of the M-English DC/TMD is a reliable and valid instrument that can be utilised among the English-speaking Malaysian population in the diagnosis of TMD.

## Figures and Tables

**Figure 1 healthcare-10-00329-f001:**
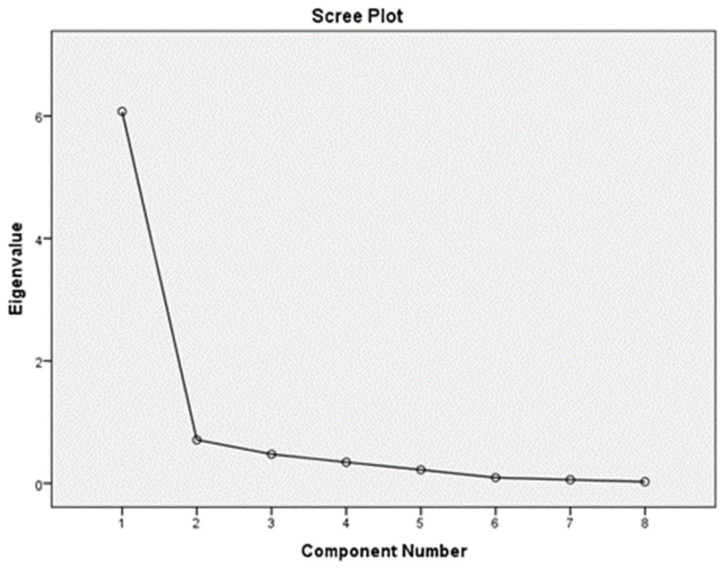
EFA of the GCPS 2.0 demonstrated a 1-factor structure scree plot.

**Figure 2 healthcare-10-00329-f002:**
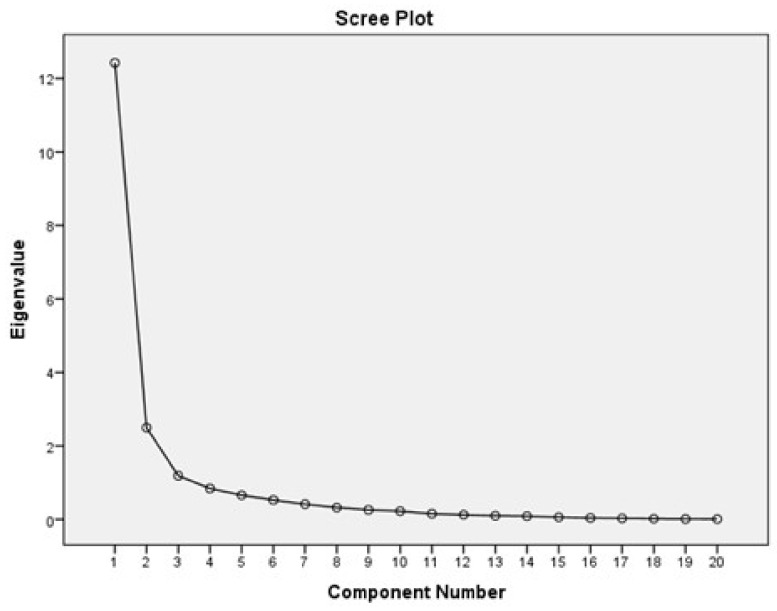
EFA of JFLS-20 demonstrated a 3-factor structure scree plot.

**Table 1 healthcare-10-00329-t001:** Descriptive statistics of the demographic data (*n* = 208).

Variable	*n* (%)
Age group/year	
18–30	112 (53.8)
31–40	44 (21.2)
41–50	24 (11.5)
51–60	8 (3.8)
61 and above	20 (9.6)
Gender	
Male	67 (32.2)
Female	141 (67.8)
Marital status	
Married	96 (46.2)
Separated/divorced	5 (2.4)
Widowed	6 (2.9)
Never married	101 (48.6)
Level of education	
Primary school	11 (5.3)
Secondary school	43 (20.7)
Diploma/College	85 (40.9)
Degree	58 (27.9)
Masters/PhD	11 (5.3)
Monthly income (MYR)	
<1200	55 (26.4)
1201–2500	46 (22.1)
2501–5000	68 (32.7)
5001–7500	26 (12.5)
>7500	13 (6.3)

**Table 2 healthcare-10-00329-t002:** Changes made after content and face validity assessment.

Domain	Item	Original English DC/TMD Question	M-English DC/TMD Question
JFLS-20	3	Chew chicken (e.g., prepared in oven)	Chew chicken (e.g., roasted chicken)
	5	Chew soft food (e.g., macaroni, canned or soft fruits, cooked vegetables, fish)	Chew soft food (e.g., kuey teow, canned or soft fruits, cooked vegetables, fish)
	6	Eat soft food requiring no chewing (e.g., mashed potato, apple sauce, pudding, pureed food)	Eat soft food requiring no chewing (e.g., mashed potato, pudding, porridge)
OBC	7	Hold or jut jaw forward or to the side	Hold or move jaw forward or to the side
	14	Play musical instrument that involves use of mouth or jaw (for example, woodwind, brass, string instruments)	Play musical instrument that involves use of mouth or jaw (for example, saxophone, flute, trumpet, viola)

**Table 3 healthcare-10-00329-t003:** Concurrent validity (Spearman’s rho correlation test) of the GCPS 2.0 and JFLS-20 domains.

Concurrent Validity	Correlation Coefficient (*r*)	*r* Value Descriptor	*p*-Value
GCPS 2.0—BPI			
“Characteristic Pain Intensity” subdomain	0.81	Strong positive	*p* < 0.01
“Interference of function” subdomain	0.75	Strong positive	
JFLS-20/OHIP-TMD			
JFLS-20/OHIP-TMD all subdomains	0.75	Strong positive	*p* < 0.01

**Table 4 healthcare-10-00329-t004:** Convergent and discriminant validity (Spearman’s rho correlation test) of the GCPS 2.0 and JFLS-20.

Convergent & Discriminant Validity	Correlation Coefficient (*r*)	*r* Value Descriptor	*p*-Value
GCPS 2.0			
OHIP-TMD “physical pain” subdomain	0.67	Moderate positive	*p* < 0.01
OHIP-TMD “psychological discomfort” subdomain	0.58	Moderate positive	
OHIP-TMD “physical disability” subdomain	0.65	Moderate positive	
OHIP-TMD “psychological disability” subdomain	0.64	Moderate positive	
OHIP-TMD “social disability” subdomain	0.70	Moderate positive	
OHIP-TMD “handicap” subdomain	0.61	Moderate positive	
GSROH score	−0.68	Moderate negative	
JFLS-20			
OHIP-TMD “functional limitation” subdomain	0.69	Moderate positive	*p* < 0.01
GSROH score	−0.86	Strong negative	

**Table 5 healthcare-10-00329-t005:** Known-group validity of GCPS 2.0 and JFLS-20 (Mann-Whitney U test).

Domain	Group (*n*)	Mann-Whitney U (U)	Mean Rank	*p*-Value
GCPS 2.0	General public (*n* = 119)	2495	80.97	*p* < 0.01
	TMD (*n* = 89)		135.97	
JFLS-20	General public (*n* = 119)	2065.5	77.36	*p* < 0.01
	TMD (*n* = 89)		140.79	

**Table 6 healthcare-10-00329-t006:** Test-retest reliability of GCPS 2.0 and JFLS-20 (Intraclass Correlation Coefficient).

Test-Retest (ICC) (*n* = 50)	Single Measures	95% Confidence Interval
GCPS 2.0		
Total score	0.95	0.91–0.97
“Characteristic Pain Intensity” subdomain	0.97	0.94–0.98
“Interference of function” subdomain	0.92	0.85–0.95
“Days with interference” subdomain	0.92	0.87–0.95
JFLS-20		
Total score	0.97	0.94–0.98
“Mastication” subdomain	0.98	0.96–0.99
“Vertical jaw mobility” subdomain	0.97	0.95–0.98
“Verbal and non-verbal communication” subdomain	0.92	0.87–0.95

## Data Availability

The datasets used and/or analysed during the current study are available from the corresponding author on reasonable request.

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
