# Peer review of "Reliability and Validity of the Malaysian English Version of the Diagnostic Criteria for Temporomandibular Disorder (M-English DC/TMD)"

_healthcare, 2022, doi:10.3390/healthcare10020329_

Round 1

Reviewer 1 Report

The gender distribution of the sample is not homogeneous despite the fact that the sample was selected for convenience.

Author Response

This research describes the cross-cultural adaptation of Malaysian English version of DC/TMD where females suffered from the condition more than male with 2:1 ratio. Therefore, more females were recruited in this research which were similar to other research:

  1. National Institute of Dental and Craniofacial Research. (2018). Retrieved from https://www.nidcr.nih.gov/research/datastatistics/facialpain/prevalence
  2. Ohrbach R: Dworkin SF: The Evolution of TMD Diagnosis: Past, Present, Future. J Dent Res 2016, 95:1093-1101.
  3. Johansson A, et al: Gender difference in symptoms related to temporomandibular disorders in a population of 50-year-old subjects. J OROFAC PAIN 2003; 17:29-35.

Reviewer 2 Report

Comments on ”Reliability & Validity of the Malaysian English version….”

A nice and informative introduction!

Under material and methods much is declared but not explained, a little bit more is explained under Discussion. The statistics are difficult and if you address statistical experts it might be enough but if you want to explain to clinicians you ought to try to explain the statistical analysis better. Some parts of the discussion could be moved to the M&M.

I appreciate your discussion about some items of the indexes regarding very private things, I have experienced the same reaction from patients in my country

Author Response

Thank you for your comments. We have revised the manuscript according to your suggestion & please refer to:

  1. Page 3, Methods, Line 94-99
  2. Page 4, Methods, Line 183-184
  3. Page 4, Methods, Line 192-194
  4. Page 5, Methods, Line 204-208
  5. Page 10, Discussion, Line 301-306

Thank you for sharing your experience with us.

Reviewer 3 Report

The article is definitely too long and. I propose to shorten it absolutely. This is not intended to be a report on the work carried out, but a scientific article that is easy for the reader to read. Now, a lot of redundant descriptions are absent.

Author Response

Thank you for your comment. We have reviewed the manuscript and shortened it where possible especially the discussion part. Thank you.
